# Typing tumors using pathways selected by somatic evolution

Sheng Wang[1], Jianzhu Ma [2,3], Wei Zhang[2,3], John Paul Shen[2,3,4], Justin Huang[2,3,5], Jian Peng[1,6] & Trey Ideker [2,3,4,5]

Many recent efforts to analyze cancer genomes involve aggregation of mutations within reference maps of molecular pathways and protein networks. Here, we find these pathway studies are impeded by molecular interactions that are functionally irrelevant to cancer or the patient's tumor type, as these interactions diminish the contrast of driver pathways relative to individual frequently mutated genes. This problem can be addressed by creating stringent tumor-specific networks of biophysical protein interactions, identified by signatures of epistatic selection during tumor evolution. Using such an evolutionarily selected pathway (ESP) map, we analyze the major cancer genome atlases to derive a hierarchical classification of tumor subtypes linked to characteristic mutated pathways. These pathways are clinically prognostic and predictive, including the *TP53-AXIN-ARHGEF17* combination in liver and *CYLC2-STK11-STK11IP* in lung cancer, which we validate in independent cohorts. This ESP framework substantially improves the definition of cancer pathways and subtypes from tumor genome data.

[1] Department of Computer Science, University of Illinois at Urbana-Champaign, Urbana, IL 61801, USA. [2] Department of Medicine, University of California San Diego, La Jolla, CA 92093, USA. [3] The Cancer Cell Map Initiative (CCMI), La Jolla and San Francisco, La Jolla, CA 92093, USA. [4] Moores UCSD Cancer Center, La Jolla, CA 92093, USA. [5] Bioinformatics and Systems Biology Program, University of California San Diego, La Jolla, CA 92093, USA. [6] Carle Illinois College of Medicine, University of Illinois at Urbana-Champaign, Urbana, IL 61801, USA. These authors contributed equally: Sheng Wang, Jianzhu Ma. Correspondence and requests for materials should be addressed to J.P. (email: jianpeng@illinois.edu) or to T.I. (email: tideker@ucsd.edu)

One of the most striking findings of the cancer genome sequencing projects has been the extreme heterogeneity in genetic alterations observed among tumors[1–3]. Each new tumor genome that is sequenced presents a new collection of genetic mutations that have, save for a few recurrent events, been only rarely observed before. This heterogeneity poses a fundamental challenge to efforts to understand and treat cancer, since such efforts largely depend on finding recurrent patterns in data.

Among the ongoing attempts to address cancer heterogeneity, an important paradigm has been to aggregate gene mutations into higher level structures and functions in cancer cells, such as protein complexes, signaling pathways, and biological processes. Such pathway analyses have been frequently applied to cancer datasets to aggregate gene-level signals to identify new pathway-level biomarkers[4–7], to increase sensitivity for identification of cancer driver genes[8,9], and to discover key regulators of cancer-related transcription[10,11]. Moreover, different genetic alterations perturbing the same cancer pathway are found to drive the same, or similar, cancer subtypes and associated clinical outcomes[9].

Methodologically, many approaches to cancer pathway analysis have been based on aggregating mutations across neighboring genes in a network of previously defined molecular interactions[4,12–16]. A popular model is heat diffusion, also called network propagation[17], by which individual gene mutations in a tumor are diffused, like sources of heat, across the network. Such diffusion creates "hot" network neighborhoods of genes proximal to mutated genes. These network neighborhoods define cancer driver pathways[4,7] and potential drug targets for cancer therapy[18–20]. They also allow patients to be clustered into subtypes, because the neighborhoods, unlike individual genes, are commonly mutated and thus provide a basis for grouping tumors[9,21]. Other than network propagation, related methods include network clustering[22], network integration[23], and network regularization[9].

Ideally, such pathway analyses should rely on the specific molecular interactions that drive cancer in relevant tissue types, as opposed to interactions important for other cellular states, diseases and/or tissues. However, most types of experimental data used to inform molecular interaction networks, including protein–protein interactions and genetic interactions, cannot yet be readily generated at the scale necessary to cover many specific tumor samples or tissues. Therefore, in nearly all cancer pathway analyses, molecular interaction information is drawn heavily from network meta-resources[7–9]. These meta-resources are large, cataloging in the range of $10^3$–$10^7$ interactions, as well as non-discriminatory, representing many diverse experiments in different human cell lines, primary tissues, or ex-vivo contexts such as yeast two-hybrid[24], with each source influenced by different rates of false-positive and false-negative errors.

While these meta-resources have been extremely useful, the high diversity of their contents motivates at least two major directions for further bioinformatics research. First, the effects of large numbers of non-specific interactions are not yet well understood. Is their inclusion in cancer pathway analyses helpful, neutral, or harmful? Second, it is not yet clear how to formulate molecular interaction networks that are both cancer-relevant and tissue-type specific. While various computational methods have been proposed to address tissue specificity, for instance by selecting interactions with tissue-specific gene expression patterns or functional annotations[15,25], similar strategies have not been devised for nominating interactions specific or relevant to cancer.

Here we show that, in fact, the informative pathways driving cancer pathogenesis and subtypes can be remarkably difficult to identify in the presence of many gene interactions irrelevant to cancer. We find that this problem can be at least partially addressed by creating a stringent filter on molecular interaction resources, based on patterns of mutually exclusive genetic alterations which arise during tumor evolution[7,26]. We use the resulting cancer- and tissue-specific network, which we call the Evolutionarily Selected Pathway map, to analyze tumor genomes from The Cancer Genome Atlas and International Cancer Genome Consortium, resulting in a taxonomy of cancer pathways and subtypes associated with clinical outcomes.

## Results

**Random interactions diminish the influence of pathways**. To explore the effects of irrelevant gene interactions on cancer pathway analysis, we first simulated a somatic mutation dataset consisting of a gene-by-tumor matrix of binary mutated/unmutated states for each gene across multiple tumors. Mutations were generated randomly considering the existence of frequently mutated pathways (FMP) and frequently mutated genes (FMG), both of which were mutated with equal elevated probability in a tumor relative to the remaining background genes (Fig. 1a). In mutating FMPs, mutations were assigned to a fixed number of member genes in the pathway. The simulated tumors were then annotated to subtypes according to the status of a selected FMP, i.e., tumors for which this pathway was mutated were assigned FMP subtype 1 and otherwise FMP subtype 2 (Fig. 1a). As an alternative, we also considered subtype assignments following an FMG rather than a pathway.

Next, we sought to determine how well these simulated tumor subtypes could be recovered by current pathway analysis approaches. Following a standard methodology[27], the mutation profile of each tumor was propagated across a gene interaction network, which we constructed by densely connecting sets of genes representing FMPs, embedded within otherwise random interactions (Fig. 1b). Random networks were simulated using an Erdos-Renyi model[28]. We found empirically that the method for generating random interactions, preferential attachment[29] or Erdos-Renyi[28], did not have a large effect on the analysis (Supplementary Figs. 1, 2 and 3). The propagated profiles for all tumors were then clustered into two groups (Fig. 1c) using the $k$-means + + algorithm[30]. The agreement between these clusters and the correct subtypes was measured using the Adjusted Rand Index (ARI). For further details of simulations, see Methods.

When the reference network contained few random interactions ( < 1%), we found that FMP-driven subtypes were recovered with very high accuracy, even in the presence of background mutations (Fig. 1d and Supplementary Fig. 2a; 95% accuracy for a model with 1% random interaction density and 1% background mutation frequency). This performance was robust to a range of interaction densities within the pathway, such that high accuracy could still be achieved with as few as 20% of interactions among pathway member genes (Fig. 1e). In contrast to pathways, the ability to recover single gene (FMG) subtypes dropped sharply as background mutations were added, consistent with previous reports that pathway analysis can boost power to detect subtypes[9] in comparison to analysis of individual gene mutations (Fig. 1d).

When increasing numbers of random interactions were added to the network (i.e., cancer irrelevant), the accuracy of FMP subtype recovery gradually decreased. For example, by increasing the random interaction density to connect 2% of gene pairs, we found that accuracy of subtype recovery fell to less than 10% (Fig. 1f and Supplementary Fig. 2b). This result raised a warning that cancer pathways might be difficult to discern within the large interaction databases commonly used for network analysis, in which interactions typically connect 1–2% of gene pairs[31].

Further exploration led to the curious observation that, even in the presence of random interactions, high accuracy could be restored by simulating FMPs only, while excluding FMGs from

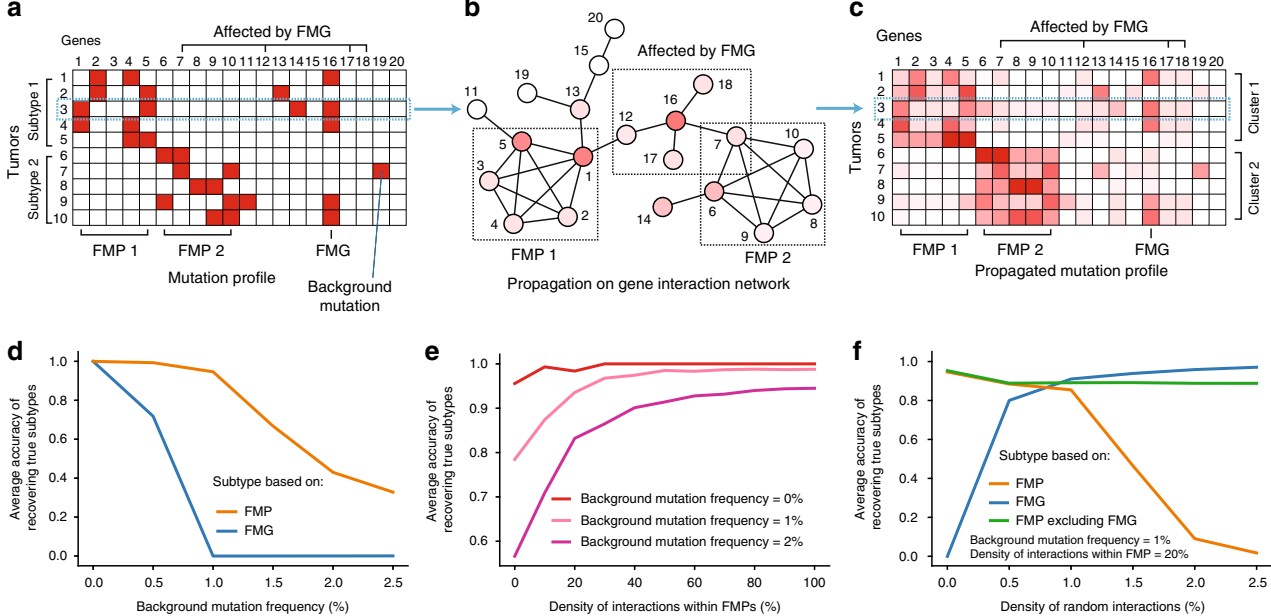

**Fig. 1** Exploring cancer pathway analysis through simulation. **a** Simulated somatic mutation dataset including two FMPs (genes 1–5) and an FMG (gene 16). Mutated genes are shown in red and non-mutated genes are shown in white. A reduced set of tumors and genes (10 × 20) is shown as an example; the full simulation is (1000 × 1000). **b** Network-based propagation of mutations over a simulated gene interaction network. Shades of red show propagated mutation values for tumor sample #3. **c** Mutation dataset from **a** following network propagation. **d** Tumor stratification performance with increasing frequency of background mutations, when no random interactions are presented. Performance is measured by calculating Adjusted Random Index (ARI) between the true subtypes and the tumor clusters derived by network stratification. **e** Tumor stratification performance with increasing density of interactions within FMPs, when no random interactions are presented. **f** Tumor stratification performance with increasing random gene interaction density

the mutation model (Fig. 1f and Supplementary Fig. 2b). Prompted by this observation, we then considered a complementary scenario in which subtypes were driven by an FMG rather than FMPs. Remarkably, the accuracy of FMG subtype recovery actually improved as random interactions were added to the network, suggesting that random interactions amplify the mutation signal of individual genes relative to pathways (Fig. 1f and Supplementary Fig. 2b). Inspection of the model revealed that this difference occurs because an FMG propagates its mutation state more readily to network neighbors: All interactions of an FMG spread mutation signal outwards, whereas only some interactions of an FMP do, with others being internal to the pathway. Therefore, in the presence of sufficient random interactions, the mutation signal of pathways is eclipsed by the mutation signal of strong individual driver genes, an outcome which runs counter to the goal of pathway analysis. These same qualitative results were seen for a range of background mutation frequencies and interaction densities (Methods, Supplementary Figs. 1 and 3).

**A stringent map of evolutionarily selected pathways**. Given that pathway analysis was adversely affected by random (irrelevant) interactions but robust to missing interactions within pathways, we sought to derive a new cancer gene interaction reference map using a very stringent policy (Fig. 2, Methods). Out of the many sources to construct such a network (e.g., strength of a certain type of data, literature, expert curation), exploratory analysis revealed two features as particularly important: biophysical interaction among gene products and epistatic genetic interaction during somatic evolution (Supplementary Figs. 4–7). Biophysical (or protein–protein) interactions define the physical architecture of cancer pathways, including protein complexes and signaling cascades. Epistatic genetic interactions connect genes in which the functional effects of genetic mutations are inter-dependent[32–34].

Mutual exclusivity, a type of epistatic interaction whereby two genes are rarely co-mutated during somatic evolution of a tumor, has been used extensively to prioritize functionally related cancer genes[26,35,36]. Combining these two features, we selected biophysical interactions from the InBioMap resource[37] in which the two genes exhibited mutual exclusivity within one or more cancer types (Methods). The resulting network of tumor type-specific interactions, which we call an Evolutionarily Selected Pathway (ESP) map, covered 258 genes and 263 interactions (Fig. 2 and Supplementary Data 1).

**Using the ESP map to stratify tumors**. Given this ESP map, we applied it to analyze somatic mutation profiles of 18 cancer cohorts in The Cancer Genome Atlas (TCGA, Methods). As above, network propagation was used to spread the influence of gene mutations in each tumor to network neighbors, along only those interactions supported by mutual exclusivity in the corresponding tumor tissue. The propagated tumor mutation profiles for each tissue were compressed into low dimensions and clustered into subtypes of increasing resolution ($k = 2\ldots6$) using standard consensus clustering with the $k$-means $++$ algorithm[30] (Fig. 3a, b). By contrasting the propagated mutation profiles across subtypes of cancer patients, each subtype was assigned one or more network regions—which we call characteristic ESPs— that were impacted by mutations enriched in that subtype (Fig. 3c, Methods).

For example, this procedure stratified 430 HPV-negative head and neck squamous carcinomas (HNSC) into a hierarchy of subtypes (Fig. 4a) that were strongly associated with patient survival (Fig. 4b–d) and clinical variables including smoking status and recurrence (Fig. 4a bottom, and Supplementary Data 3). The characteristic ESPs for these subtypes included genes well-known to function in pathogenesis of head and neck cancer (Fig. 4e, TP53, NOTCH, CDKN2A) as well as many genes

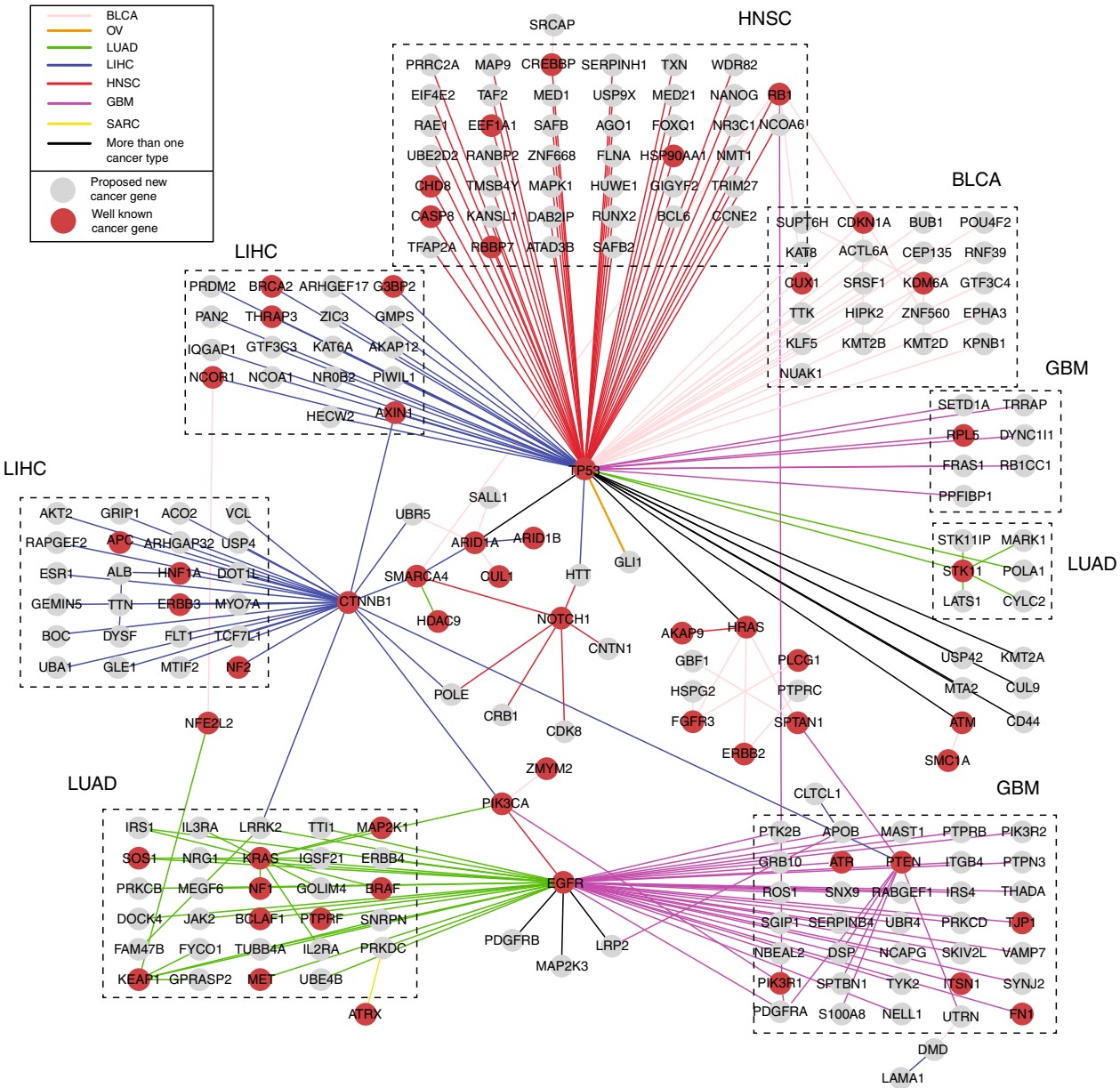

**Fig. 2** Evolutionarily selected pathway (ESP) map. All interactions in the map are supported by prior evidence of both biophysical protein–protein interaction and mutual exclusivity of mutation during somatic evolution. Interaction colors represent the cancer type in which mutual exclusivity is observed. Black edges indicate interactions for multiple cancer types (Supplementary Data 2). Boxes indicate interaction neighborhoods enriched in specific cancer types. The largest connected component of ESP map is shown, covering 88% of the complete map (226/256 genes) and focused on seven cancer types in which the ESP map leads to superior subtype stratification (see text): BLCA bladder cancer, GBM glioblastoma, HNSC head and neck squamous carcinoma, LIHC liver hepatocellular carcinoma, LUAD lung adenocarcinoma, OV ovarian cancer, SARC sarcoma

that had not been previously described to have roles in this disease. Subtype hierarchies for other tumor tissue types are provided in the supplement (Supplementary Figs. 8–14).

**Exploration and validation of characteristic ESPs**. To systematically evaluate the stratification results, we first compared the ESP subtypes to previously annotated cancer subtypes and clinical variables for each tissue as recorded by TCGA. Indeed, some ESP subtypes closely tracked known clinically identified subtypes (Supplementary Data 3). For instance in breast cancer, ESP subtypes were significantly correlated with Estrogen Receptor and *HER2* expression status, while in uterine cancer, subtypes were significantly correlated with the histological subtype, serous vs.

endometrioid. In colorectal cancer, ESP subtypes were correlated with *KRAS* and *BRAF* mutation status and also separated primary colon from primary rectal tumors. Notably, these distinct origins had been considered indistinguishable from analysis of somatic mutation profiles alone[38] (this previous analysis was in the absence of molecular network information). In other cases, including head and neck, liver, and bladder, the resulting stratification of tumors corresponded only weakly to known clinical subtypes and variables, or not at all, suggesting new disease subtypes and pathways worthy of further investigation (Supplementary Data 3).

We next examined the ability of the ESP subtypes to stratify patients according to progression-free survival time, a common quantitative means of assessing the utility of subtype

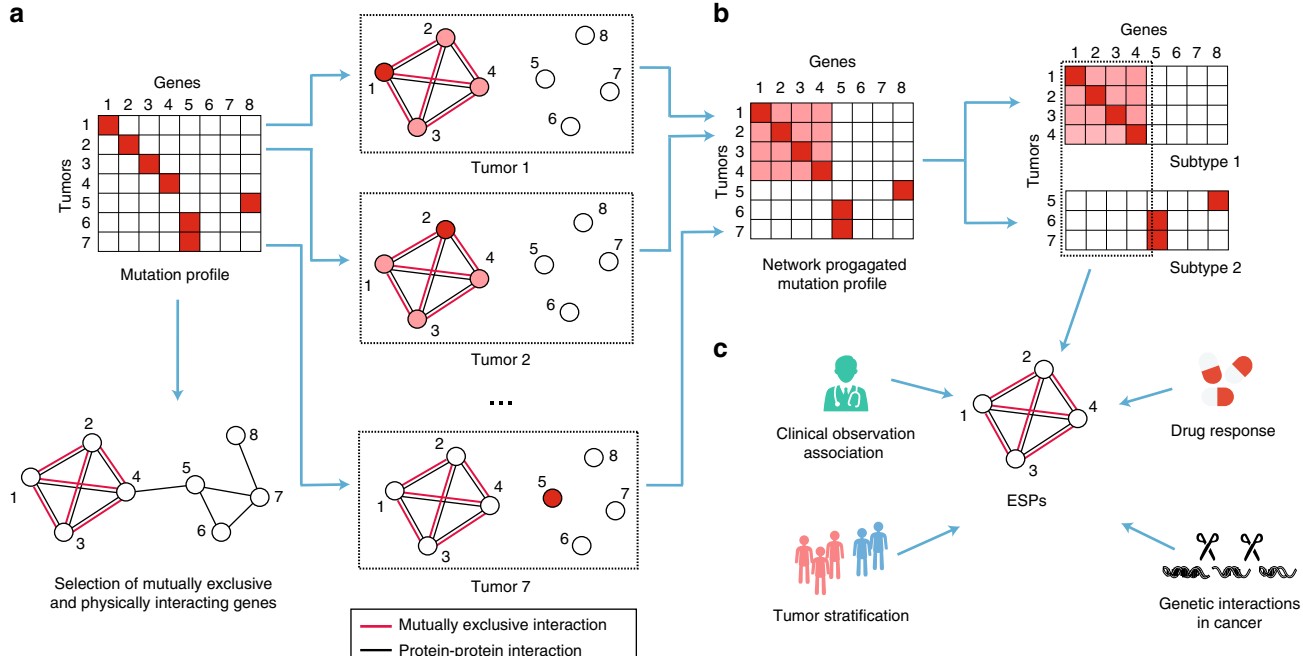

**Fig. 3** Flowchart of ESP stratification. **a** Selection of mutually exclusive protein–protein interactions and propagation of tumor somatic mutation profiles. Color scheme as for Fig. 1. **b** Clustering of network-propagated mutation profiles. **c** Detection of characteristic ESPs by comparing the propagated mutation profiles between subtypes

stratification[9,39]. We found that 8 of the 18 tumor cancer cohorts could be stratified into prognostic subtypes, i.e., which were significantly associated with differences in patient survival ($p < 0.01$, BLCA, GBM, HNSC, LIHC, LUAD, OV, SARC, THCA). This prognostic ability was quite favorable compared to standard clustering of somatic mutation profiles in the absence of network information (Fig. 5a). Generally, with few exceptions it was also favorable in comparison to the prognostic value of known subtypes and clinical variables (Supplementary Data 3). These substantial associations with patient survival provide clinical support to the subtypes identified by the ESP mapping procedure.

Next, we studied whether mutation of the identified ESPs leads to downstream effects on gene expression. Indeed, the alteration status (mutated or unmutated) of three ESPs, THCA_551, LUAD_531, and SARC_531 (Supplementary Data 1), was associated with the first principal component of expression among tumor samples in thyroid carcinoma, lung, and sarcoma, respectively (Supplementary Data 4). More specifically, we checked the influence of ESP alteration on the expression of known cellular pathways associated with cancer (Methods). In this case, we found a larger proportion of ESPs (82 out of 117) were associated with expression changes in at least one of these specific cancer pathways.

Further support was provided by a number of exploratory analyses, of which we mention the most important findings here (Supplementary Figs. 4–7). First, we compared different ESP maps in which biophysical protein interactions were drawn not from InBioMap but from STRING[40] or PathwayCommons[41], alternative database sources frequently used in cancer pathway analyses. Of these sources, InBioMap had the best ability to stratify patients by survival (Supplementary Fig. 7). Second, we compared ESP map against the much larger complete set of 606,195 protein interactions in InBioMap, unconstrained by epistatic genetic interactions. As anticipated by our earlier simulations, this large network did not find prognostic subtypes for any of the 18 cancer cohorts examined (Fig. 5b). Analysis with

ESP map also yielded a higher density of known cancer driver genes than analysis with the full InBioMap (Supplementary Fig. 15). Third, we evaluated the complementary configuration: a network constructed solely by mutual exclusivity of gene pairs, unconstrained by biophysical interactions. For most of the evaluated cancer types, the performance was not as good as that of the ESP map, indicating that epistatic interactions are more informative when combined with biophysical interactions (Supplementary Figs. 4–6). Fourth, we examined the impact of total network size by progressively adding biophysical interactions to the ESP map based on their significance of mutual exclusivity (Methods). For most cancer types, we saw the best stratification performance (survival association) with the top ~100 mutually exclusive gene interactions. Remarkably, adding further interactions led to a drop of performance that was seen consistently across tissues (Fig. 5c). Collectively these analyses led us to select a reference network approximately three orders of magnitude smaller than the very large networks used in previous cancer studies (Fig. 2, 263 interactions in ESP vs. > 10^5 interactions for unconstrained interaction databases such as InBioMap, STRING, or PathwayCommons)[4,7,9].

**A *TP53-AXIN-ARHGEF17* pathway associated with liver cancer**. As one particular case study, we examined an ESP subtype associated with poor survival in liver cancer which was mutated in 36% of tumors (Fig. 6a, b). *AXIN1* and *TP53* form a previously described protein complex, in which *AXIN1* phosphorylates *TP53* in response to DNA damage, triggering cell-cycle arrest or apoptosis[42]. In contrast, the biophysical interaction between *ARHGEF17* and *TP53* had not been studied in depth, having been one of many interactions detected in a large-scale interaction screen[43]. Mutations to *ARHGEF17* and *TP53* were mutually exclusive within this subtype and this subtype only (p-value = 0.05 using one tailed Fisher's exact test, p-value = 0.21 for the liver tumor cohort at large), supporting the biophysical interaction and suggesting it is subtype-specific.

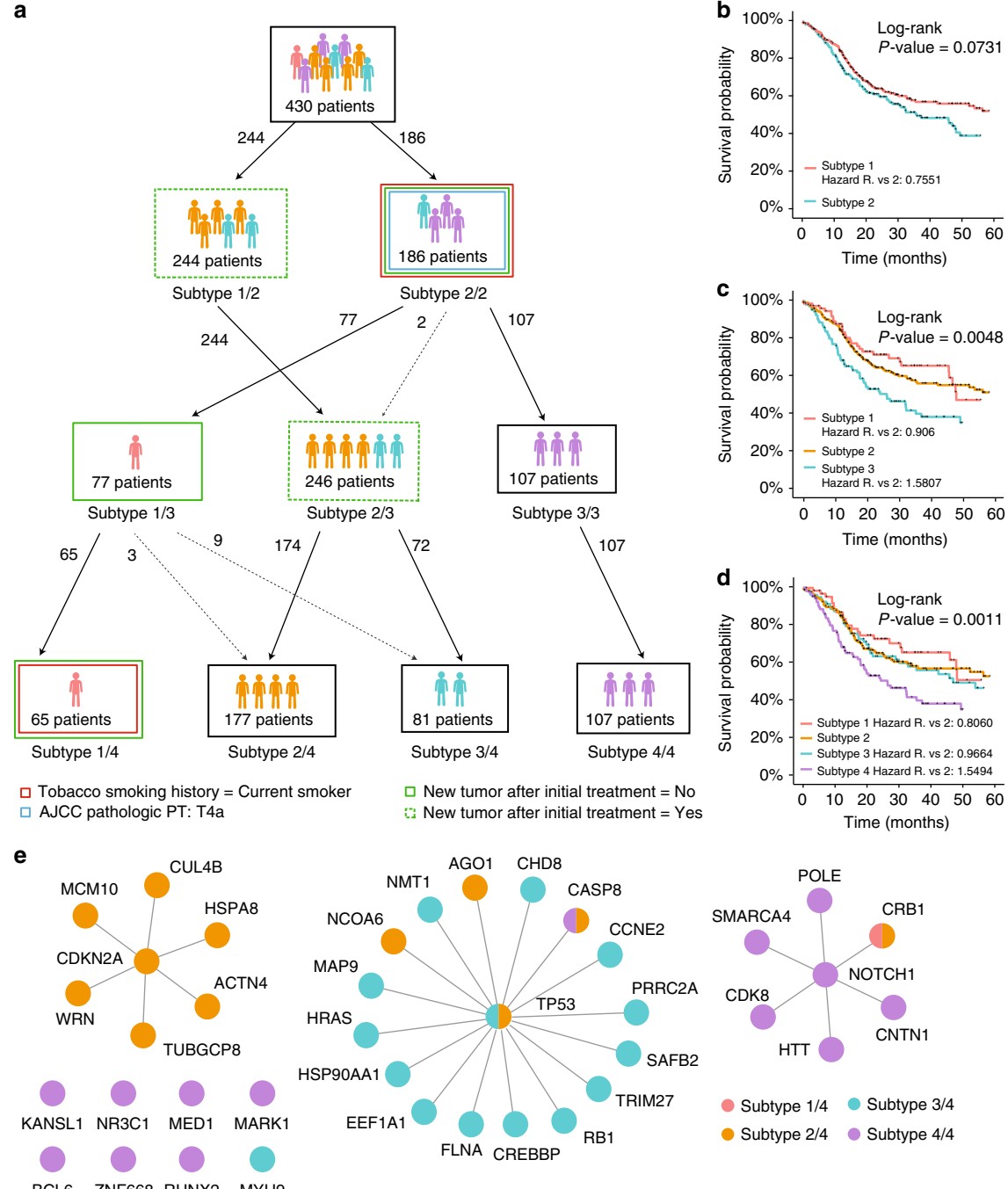

**Fig. 4** Hierarchical stratification and characteristic ESPs of head and neck cancer. **a** Hierarchical organization of tumors at increasing subtype resolution. Layer $k$ of the hierarchy corresponds to clustering cancer patients into $k$ subtypes ($k = 1,2,3,4$). **b–d** Kaplan–Meier survival plots when stratifying tumors into two (**b**), three (**c**), or four (**d**) subtypes. **e** Mutated ESPs used in assortment of patients into four subtypes

We identified two interesting properties relating to genes in this ESP that had not been reported by previous studies. First, mutating this ESP indicated low survival, which could be further validated in a second large cohort of liver tumors from the International Cancer Genome Consortium (ICGC), supporting the value of this ESP as a putative clinical biomarker (Hazard ratio 2.7, Fig. 6c). Notably, none of the three genes involved in this ESP (*TP53*, *AXIN1*, and *ARHGEF17*) was significantly prognostic when mutations to each gene were considered individually (Fig. 6d). Second, given that the *AXIN1-TP53* interaction modulates DNA damage response, we also tested associations between this ESP and DNA damaging

chemotherapeutics widely used to treat liver cancer (Methods). This investigation was carried out in a panel of 19 liver cancer cell lines characterized in the Genomics of Drug Sensitivity in Cancer (GDSC) dataset, of which 15 had mutations placing them in the ESP subtype. This analysis showed that ESP subtype mutations are indeed associated with strong resistance to mitomycin C (Fig. 6e and Supplementary Fig. 16a), consistent with the observation of poor survival in both TCGA and ICGC (Fig. 6b, c).

**A *CYLC2-STK11-STK11IP* pathway associated with lung cancer.** As a second case study, we examined an ESP associated with aggressive lung cancer, mutated in 22% of lung tumors and

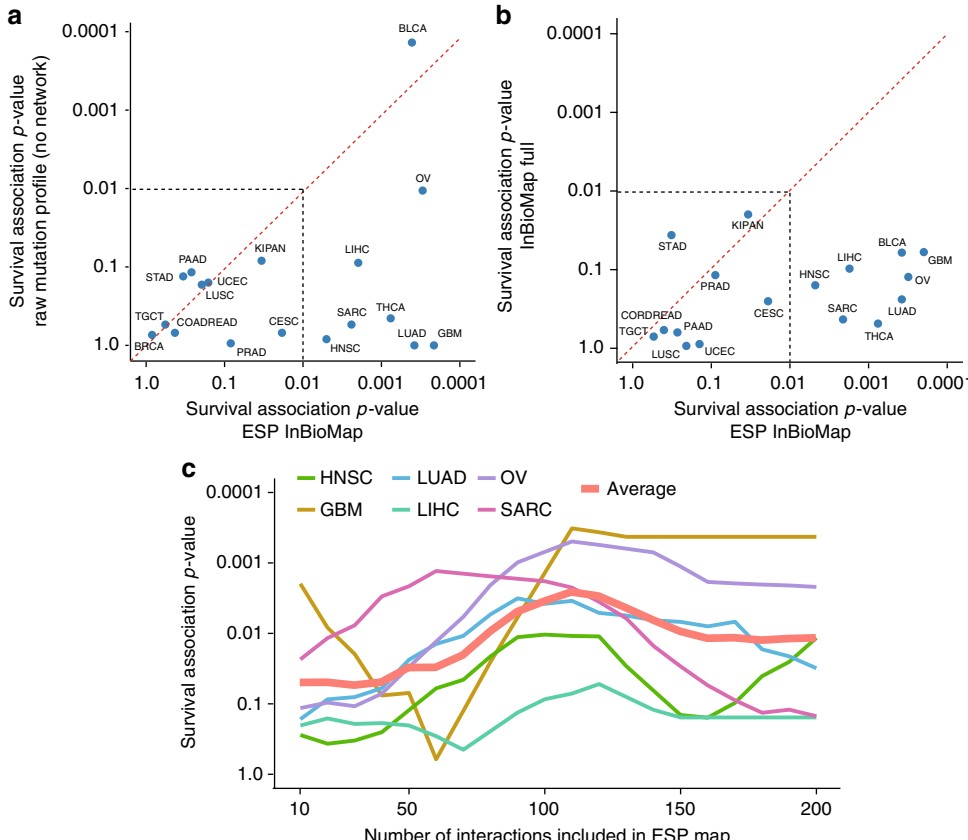

**Fig. 5** Evaluation performance of different input sources. For each method, performance was evaluated as the prognostic ability of the resulting subtypes, i.e., the association of subtype with patient overall survival computed by log-rank *p*-value of Cox regression. **a** ESP of InBioMap vs. "Empty" network. **b** ESP of InBioMap vs. Complete InBioMap. **c** Performance as the threshold number of Mutually Exclusive gene interactions increases from 10 to 200

involving mutations to the genes *CYLC2*, *STK11* and *STK11P* (Fig. 6f). As validation, we found that patients of this ESP subtype had significantly lower survival not only in the original TCGA cohort we examined (Fig. 6g) but also the MSK-IMPACT cohort (Fig. 6h). As we had observed for liver, we found that the prognostic significance of this lung ESP was not tied to any individual gene but required the integration of mutations across the pathway (Fig. 6i). Lung cell lines mutated in this ESP were significantly less sensitive to paclitaxel, a common chemotherapy in treatment of lung cancer (Fig. 6j and Supplementary Fig. 16b), providing a rationale for the observed poor survival in patients.

Among the three genes, *STK11* stood out as the only gene well-studied in this cancer type[44–46], in which *STK11* mutation has been associated with decreased immune surveillance and lack of response to immune checkpoint inhibitors. The biophysical interaction between *STK11* and *STK11IP* had not been directly studied in cancer but is known to play a role in Peutz–Jeghers syndrome, in which patients are at high risk to further develop cancers of multiple types[47]. The physical and epistatic interactions between *STK11IP* and *CYCL2* had not been previously studied.

## Discussion

Pathways have been extensively applied in cancer genetics to organize cancer driver genes and stratify patients into subtypes[7–9,31]. Since interaction mapping techniques do not yet generate comprehensive datasets tailored to each specific tumor and tissue in a cohort, pathway analyses typically strike a compromise, by pooling molecular interactions derived from many

previous experiments into one large protein network. Here, we have demonstrated that this compromise can be a major limitation, as it dilutes the signal of pathways relative to frequently mutated genes. We showed that this problem could be addressed by reinforcing biophysical interactions with epistatic genetic interactions observed directly in populations of tumors, leading to creation of a tissue-specific resource, the ESP map, of broad use in the study of cancer pathways and subtypes.

The general findings of our study may have relevance to a spectrum of network analysis methods in current use. Several popular tools for identifying cancer pathways are based on the technique of network propagation, as we have used here, including HotNet2, TieDie, Paradigm, Network-Based Stratification, and NetSig, among others[7–9,48,49]. More broadly, any method that examines the interaction partners of a gene to identify disease genes and pathways may be adversely affected when mining large reference interaction networks containing disease-irrelevant interactions. Such concerns are not restricted to the field of cancer genetics but likely apply to other diseases and end-goals, such as annotation of gene function, a field replete with network-based methods[50,51]. Where problems are indeed identified, there may be significant opportunities to apply some of the lessons learned here, namely, much greater stringency and specificity in reference molecular networks.

Although the present analysis has focused on identification of cancer pathways impacted by somatic coding mutations, it is important to note that such pathways likely capture just one facet of the molecular mechanisms contributing to tumorigenesis. Many other data types can reveal pathways and stratify tumors into clinically meaningful subtypes, including copy number

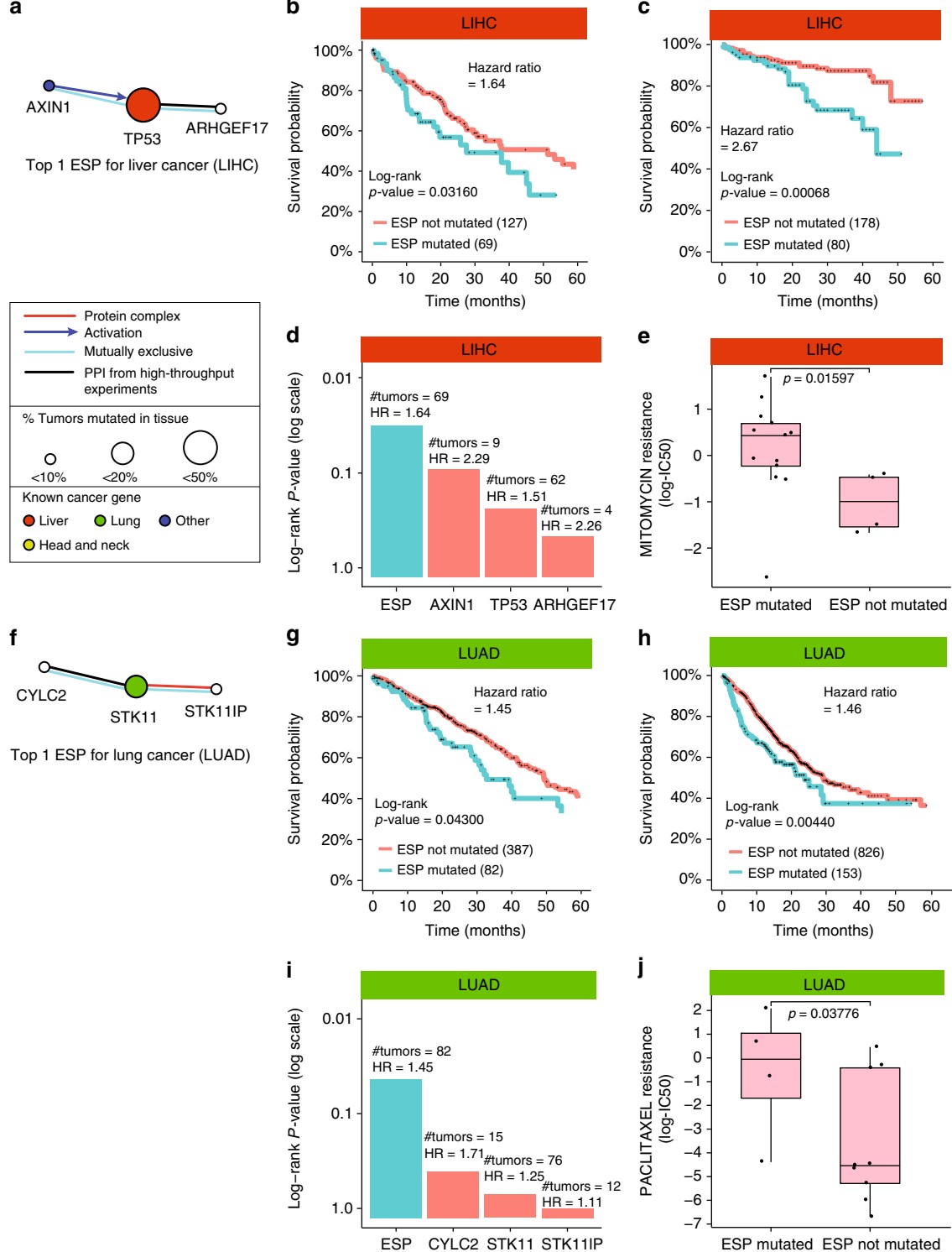

**Fig. 6** Exploration and validation of mutated ESPs in multiple cancers. **a** Mutated ESP characteristic of liver cancer subtype 1/2. **b** Kaplan–Meier survival plot for the liver ESP shown in **a** on patients in TCGA (Methods). **c** Corresponding Kaplan–Meier survival plot for patients in ICGC. **d** Stratification of patients using the aggregate of genes in the liver ESP vs. each of its genes individually. **e** Difference in drug resistance based on status of the ESP shown in **a** (Methods). Drug resistance is assessed by log(IC50) of liver cell lines in the GDSC dataset. *p*-value is calculated using a one-sided Wilcoxon signed-rank test. Box plots show the median, the 25th and 75th percentiles. **f** Mutated ESP characteristic of lung cancer subtype 1/5. **g** Kaplan–Meier survival plot for the lung ESP shown in **f** on patients in TCGA (Methods). **h** Corresponding Kaplan–Meier survival plot for patients in MSK-IMPACT. **i** Stratification of patients using the aggregate of genes in the lung ESP vs. each of its genes individually. **j** Difference in drug resistance based on status of the ESP shown in **f** (Methods). Drug resistance and *p*-value are calculated the same as for (**e**)

variants[52], noncoding somatic mutations[53], germline variants[54], and gene expression and epigenetics[55]. While it will be important to develop methods to simultaneously integrate all of these various layers (some encouraging attempts exist already[56,57]), it is also critical to understand as deeply as possible the biological information embedded in each data type individually, as we have done here for somatic coding mutations.

Looking to the future, we see many opportunities to improve upon the concept of selecting cancer-relevant gene interactions. First, algorithms for detection of single cancer genes could be applied to help identify cancer-related network regions, such as MutSigCV[58], OncodriveFM[59], ActiveDriver[60], and NetSig[8]. Second, cancer-specific molecular networks are being accumulated by different groups in increasing numbers, including networks of protein–protein interactions[61], genetic interactions[62–64], and gene-drug interactions[65–67]; one would be expect these new networks to present greater enrichment for cancer-relevant interactions, and this potential can certainly be tested. Other efforts have attempted to predict somatic mutations that are likely to perturb cell signaling[68,69], representing another promising way to identify cancer-specific networks.

## Methods

**Processing of patient mutation profiles.** As our primary discovery dataset, we downloaded somatic mutation profiles of tumors from TCGA based on whole-exome sequencing data from the GDAC Firehose website (http://gdac.broadinstitute.org, 11th February 2016). Each gene was classified as either wild type (0) or altered (1) in each patient, with alteration defined as any type of non-silent mutation. We excluded tumor types with less than 100 patients and patients with less than 10 mutations. This left in total 6240 patients with 882,110 mutations in 18,018 genes in 18 tumor type cohorts. As a validation set, we downloaded somatic mutation profiles from the ICGC portal (https://dcc.icgc.org/releases/current, 7th December 2016) including the following cohorts: OV-AU, BLCA-CN, GBM-CN, LICA-CN, LIAD-FR, LINC-JP, LIRI-JP, LUSC-CN, LUSC-KR, SKCA-BR, THCA-CN, and THCA-SA. Among these, OV-AU, GBM-CN, LICA-CN, LINC-JP, LIRI-JP, LUSC-KR, SKCA-BR, THCA-CN, THCA-SA had associated patient survival data. For these ICGC data, we excluded patients with less than 10 mutations and selected the following mutation types: missense variant, frameshift variant, non-conservative missense variant, initiator codon variant, and stop-gain. This left in total 1453 patients with 359,876 mutations in 17,135 genes. As an alternative validation set, we also obtained somatic mutation profiles for bladder cancer, lung cancer, skin cancer, brain cancer, liver cancer, ovary cancer, and thyroid cancer from MSK-IMPACT[2]. In contrast to TCGA and ICGC, which include mutation data for all genes, MSK-IMPACT uses targeted deep sequencing of 410 select cancer genes. We thus did not disregard any mutation or patient in MSK-IMPACT. In total, this dataset included 3485 patients with 28,323 mutations in 408 genes.

**Sources of molecular network data.** We consider three public databases widely used in cancer analyses, InBioMap[37], PathwayCommons[41], and STRING[40]. InBioMap aggregates PPIs from eight different gene orthology databases, transferring data to human protein pairs only if the majority of these databases agree on the phylogenetic relationship between two proteins in model organisms or humans. PathwayCommons includes PPIs from several pathway and interaction databases, focusing primarily on functional relationships between genes in canonical regulatory, signaling and metabolic pathways including hallmark pathways of cancer. STRING uses a Bayesian algorithm to integrate many different types of evidence for a protein–protein interactions, including literature curation, computationally predicted interactions, interactions transferred from model organisms by orthology, interactions computed from genomic features such as gene–gene fusion events, and interactions based on functional or co-expression similarity. All of the above network sources comprise both direct and indirect physical binding interactions between two proteins. All interactions were used as unweighted and undirected in our network propagation model.

**Evolutionarily selected pathway construction.** For each cancer type, we only focused on the gene pairs with protein interactions documented in the public gene interaction databases. We then selected top $k$ ($= 100$) most significant mutually exclusively mutated gene pairs ranked by $p$-value from these gene pairs. In particular, for any of two genes, we calculated the number of patients that (1) both genes are mutated; (2) the first gene is mutated; (3) the second gene is mutated; (4) neither of the two genes is mutated. We then calculated the $p$-value using a one tailed Fisher's exact test with these four numbers. It is possible that there is no documented gene interactions detected among these top $k$ mutually exclusive gene pairs. Then the problem degenerates to stratifying patients without using network information and thus no network propagation was performed.

**Network propagation.** For each cancer type, we mapped the mutation profile of each patient to the corresponding cancer-specific molecular network. We then propagated the mapped mutation profile to "smooth" the mutation signal across the network. Formally, let $A$ denote the adjacency matrix of a molecular network with $n$ genes. Define a gene-by-gene matrix $B$ in which each entry $B_{i,j}$ represents the probability of a transition from node $i$ to $j$:

$$B_{i,j} = \frac{A_{i,j}}{\Sigma_j A_{i,j}} \tag{1}$$

Next, let $F_t$ be a patient-by-gene smoothed mutation profile matrix. We used the random walk (with restart) algorithm to calculate $F_t$ from $B$:

$$F_{t+1} = (1 - \alpha) F_t B + \alpha F_0 \tag{2}$$

where $F_0$ is a patient-by-gene matrix representing the original patient mutation profile, and $\alpha$ denotes the restart probability controlling the relative influence of global vs. local topological information during the random walk. A larger (smaller) $\alpha$ places greater (lesser) emphasis on the local structure of the network. In practice, we found the specific value of $\alpha$ had a minor effect on our results over a sizable range (0.5–0.8; in what follows $\alpha = 0.5$). The propagation function was run recursively until $F_t$ converges ($\|F_{t+1} - F_t\|_2 < 10^{-6}$). After convergence, we normalized $F_t$ such that each row sum equals to one, so that the resulting stratification was independent of the total number of mutations per patient (mutational load).

**Dimensionality reduction.** To reduce noise brought by random passenger mutations, we projected the propagated mutation profile of each patient onto a low dimensional space using truncated SVD. With SVD, we decomposed the logarithm of the propagated mutation profile $F$ obtained by Eq. (2) into three matrices $U, S$, and $V$:

$$\log(F + c) = USV^T \tag{3}$$

where $U$ was a left-singular matrix, for which each column could be recognized as a "meta-patient" representing a group of patients with similar propagated mutation profiles. $S$ is a diagonal matrix of singular values. $V$ is a right-singular gene-by-gene matrix, for which each column can be recognized a "metagene" representing a group of genes mutated in similar patients. The value $c$ is a small positive constant (reciprocal of the number of genes) added to each entry of $F$ to avoid taking the logarithm of zero. We then truncated $U, V$, and $S$ by simply choosing the first $d$ singular vectors $U_d V_d$, and first $d$ singular values $S_d$. The projected $d$-dimention matrix $M_d$ was constructed by calculating $S_d^{1/2} U_d^T$.

**Consensus clustering.** A patient similarity matrix was constructed by calculating the cosine similarity between the columns of $M_d$ obtained by previous truncated SVD step. We adopted the $k$-means + + clustering algorithm to cluster patients using the cosine patient similarity matrix. For $k$-means ++, the maximum number of iterations was set to 100 and the number of random starts was set to 200. A consensus matrix was constructed to integrate the clustering results over different numbers of components (10–50) in SVD factorization. For each clustering result, a binary similarity matrix was constructed from the corresponding clustering labels: if two patients belong to the same cluster, their similarity is 1; otherwise the similarity is 0. A consensus matrix was calculated by averaging all similarity matrices of individual clusterings. Another $k$-means + + clustering with the same parameter setting was then applied to cluster patients using the cosine similarity between two rows of the consensus matrix. We used default $k$-means + + and truncated SVD functions implemented in Matlab.

**Simulation of somatic mutation cohorts.** We used simulations to study the function and interaction of frequently mutated pathways (FMPs) and frequently mutated genes (FMGs) to recover subtypes from somatic mutation profiles. We simulated a somatic mutation profile with 1000 tumors and 1000 genes and randomly divided the cohorts into two equal-sized subtypes. We consider three scenarios with different features to determine the tumor subtypes:

1.  Subtypes driven by mutations of FMPs: Each subtype was assigned with 25 genes in the pathway. The density of interactions within each pathway was set to 0.2 by default. For each tumor, there were two random genes mutated within each pathway. In the meanwhile, there was also an FMG in the mutation profile and it was mutated on one subtype with probability 1.0 and not mutated on the other subtype.

2.  Subtypes driven by a FMG: There was only one FMG among in the mutation profile and it was only mutated in one subtype with probability 1.0 but not the other. The overall mutation rate of the FMG is 0.5. Notice that in this setting the network structure of FMP created by scenario 1 were still in the network but it did not have any mutation enrichment.

3.  Subtypes driven by mutations of FMPs with FMG: The mutation setting on FMPs was the same as scenario 1 without any FMG presented.

We consider two types of network structure regarding the different structures of FMP and random interactions. In our first setting, we set FMP to be densely

connected subnetworks with various edge densities (from 0% to 100%) and uniformly sample $m$ (from 0% to 2.5%) random edges using an Erdos-Renyi model[28]. In the second setting, each FMP is a star-like graph and random edges are samples using preferential attachment model[29]. Besides mutations on the FMPs and the FMG, we sampled $l$ (from 0% to 2.5%) random background mutations for other genes to simulate the effect of passenger mutations in cancer. We iterated over different $l$ and $m$ to study the effect of passenger mutations and molecular interactions irrelevant to cancer during stratifying patients.

**Implementation of baseline methods.** To investigate the impact of different network sources on pathway-based tumor stratification, we compared five different kinds of input networks to ESP: 1. "empty" network with no interactions, equivalent to standard consensus clustering of somatic mutation profiles; 2. complete gene interaction network of InBioMap; 3. mutually exclusive gene pairs without considering any biophysical gene interactions; 4. Random gene network filtered by mutually exclusive gene pairs. 5. InBioMap gene interaction network filtered by known cancer pathways[56]. All these five methods were tested using the same collection of mutation profile. For methods 1, 2, 4, and 5 we used the same clustering procedure (i.e., SVD, $k$-means + + algorithm and standard consensus clustering) as discussed above. For method 3, we used the implementation of our previous work NBS[9], a network-based patient stratification method using Non-negative Matrix Factorization instead of SVD and performed network propagation using the complete gene network without applying any edge filtering. We downloaded the NBS software from its journal website[9] and used its default parameter settings. For method 2, 4, and 5 considering mutually exclusivity, we first selected top $k$ ($= 100$) mutually exclusive gene pairs without considering gene interaction information. Network propagation were only performed across these top mutually exclusive edges. For method 4, a random gene network was generated by shuffling gene name annotations. Therefore, the node degree distribution was preserved for each randomization. We also selected top $k$ ($= 100$) mutually exclusive interactions from each random PPI network and used them to perform network propagation. We repeated this process 100 times and used the average log-rank statistics to calculate the log-rank $p$-value.

**Survival analysis.** Survival analysis was performed using the R "survival" package. A Cox-proportional hazards model was fit to determine the relationship between the identified subtypes and patient survival times.

**Signature subnetwork detection algorithm.** After stratifying patients, for each subtype we identified genes that had a differentially propagated mutation score compared to the remaining cohort based on a student $t$-test. The $p$-value cutoff to determine differentially propagated genes was set at 0.05 for <4 subtypes or 0.1 for ≥4 subtypes. For each subtype, the selected set of genes was mapped back on the network to generate a subnetwork, which were taken as the characteristic ESP for the subtype.

**Validation of ESP-based patient clustering.** To validate the ability of stratifying tumors for each detected ESP, we tried to cluster tumors solely based on the selected ESP by using the same random walk-based clustering procedure was performed as before. If an individual gene was detected, then the binarized mutation profile of that gene was used to stratify tumors. Since MSK-IMPACT data only sequenced 410 cancer genes, many somatic mutations in the identified subnetworks were not included in this dataset. Therefore, when stratifying tumors in MSK-IMPACT, we did not normalize the propagated score by the total mutation numbers of each patient. Moreover, for MSK-IMPACT dataset, we used the Euclidian distance to measure the similarity between two propagated mutation profiles instead of cosine distance due to the sensitivity of cosine distance on small numbers of genes.

**GDSC cell line drug response data.** We obtained the GDSC large-scale compound response screening dataset[65], which spanned 255 chemical compounds and 990 human cancer cell lines encompassing 25 cell lineages. These 255 compounds were collected from different sources including clinical candidates, FDA approved drugs and previous documented chemosensitivity profiling experiments. To quantify drug sensitivity, we used IC50 provided by GDSC[65]. To identify the approved drugs for different types of cancer, we manually searched the useages of these drugs in the NCI database (https://www.cancer.gov/about-cancer/treatment/drugs) and Wikipedia. The drug list for each cancer type is listed in Supplementary Data 1. The somatic mutation profile and corresponding tissue of origin of these 990 cancer cell lines were also downloaded[70]. We applied our ESP-based propagation algorithm to stratify each cancer cell line into two clusters, using only subnetwork information. A rank-based Wilcoxon-type statistic was then adopted to compare the difference in drug responses of these two clusters.

**Expression association.** We clustered tumors based on a selected ESP by using a random walk-based clustering procedure described in the previous section. We then evaluated whether the expression level of a certain set of genes was significantly different for ESP-mutated and ESP-non-mutated cohorts using a one tail ranksum test. The expression level of a set of genes was represented by the first principal component of the patient-by-gene expression matrix[56].

## Data availability

A software implementation is available on GitHub at https://github.com/wangshenguiuc/NBS-ESP. All datasets used in this manuscript are available in public repositories and references are given in the text (see Processing of patient mutation profiles subsection).

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

## Acknowledgements

We gratefully acknowledge support provided by grants from the National Cancer Institutes (U54CA209891) and the National Institute of General Medical Sciences (P41GM103504 and R01HG009979) to T.I. J.P. is funded by a Sloan Research Fellowship, PhRMA Foundation Award in Informatics, and NSF Career Award (1652815). S.W. is supported by grant 1U54GM114838 awarded by NIGMS through funds provided by the trans-NIH Big Data to Knowledge (BD2K) initiative (www.bd2k.nih.gov).

## Author contributions

S.W. and J.M designed the study and developed the conceptual ideas. S.W. implemented the main algorithm. J.M. and W.Z. collected all the input sources. S.W., J.M., J.P., and T.I. wrote the manuscript with suggestions from the other authors.
