## [Peer Review File · Nature Communications]

Reviewers' comments:

Reviewer #1 (Remarks to the Author):

In this manuscript Wang et al. describe a method which uses protein-protein interactions and human cancer mutational data to generate hypotheses of which genes are in the same pathways. Some of the ESPs that they uncover predict patient survival better than any of the single genes. My comments are those of a general cancer biologist who is interested in these sort of studies to uncover novel functional tumor suppressor pathways.

Major points

1. Several aspects of this manuscript made it frustrating to review. The supposed rationale of this approach was that current approaches use protein-protein interactions that were uncovered in unrelated tissues or try to merge mutations data across cancer types. But then they go and use a protein-protein interaction dataset that is also not from a relevant cell type (they use the same protein-protein data for all cancer types). So, these protein-protein interactions are useful?
2. Figure 2 was confusing, why are some interactions boxed with the cancer name. Presumably many of those interactions were found in more than one cancer type. Even for the colored lines, if these indicate a cancer in which mutual exclusivity was observed shouldn't some interactions have two or more colored lines?
3. I am not sure the logic of looking at survival of ESPmut versus ESP nonmut patients. If mutation of the different genes in the ESP do essentially the same thing then it would have been more compelling to show that the gene expression states of tumors with each gene in the ESP look more like each other than the ESP nonmut samples. Also, even if you did look at survival with the prediction that each gene in the ESP does that same thing, then the effect on survival should have also been the same and it wasn't.
4. While they uniquely uncover ESPs with their approach are these genes really in the same pathways and relevant to cancer? What data (other than survival which can be driven by many things) do they provide that they have found any "real" interactions?

Minor points:

1. Some references do not show they claim that they do, for example ref 4-7 references in introduction are not all about cancer data sets. In the discussion, they state "...the biophysical interaction between STK11 and STK11IP had not been directly studied in cancer but is known to play a role in PJS..." but the reference that is supposed to support this does not mention STK11IP. Clearly I am not going to check all of their references but they should learn that papers need to be referenced properly and you cannot try to support a claim with a reference that doesn't support it.
2. Many terms that they used were at least somewhat wrong which made the manuscript frustrating to read, for example "18 tissue cohorts", "subtypes of a tissue" "within one or more tumor tissues". These are all from one page.
3. Some figures are also labeled in a way that makes them more confusing than they need to be. Example, Figure 6d why does that say ESP-LIHC on the top when it is ESP as well as each gene separately?
4. Figure 6e for the ESP mutated, what does that look like for each single gene (for example p53 mutant tumor would be expected to be resistant). For 6D+I they make the point that the ESP is better than each single gene but then for 6E+J then don't show the data for each individual gene.
5. For the individual genes shown in 6D and 6I isn't the p-value affected by the number of samples? What is the HR for each comparison.

Reviewer #2 (Remarks to the Author):

In this paper, the authors develop a new method for cancer stratification. In a nutshell, the method uses a diffusion process in a network referred to as Evolutionary Selected Pathways map

(ESP). ESP combines mutual exclusivity relationship and protein-protein interaction network. The underlying assumption is that if two genes are known to interact in some context and then if the mutations in these genes are mutually exclusive then the interaction is likely to be "relevant" for the particular cancer type.

With these definitions at hand, they start with the assumption that cancer subtypes can be stratified with the help of subtype specific highly mutated genes or pathways that could be discovered using a diffusion process. Considering that PPI data is noisy and that mutation data contains, in addition to driver mutations, passenger mutations the authors used simulations to test what is more problematic for the diffusion based method: inclusion of non-relevant edges (modelled as random edges) or passenger mutations (modelled as random mutations). From their simulations, they concluded that it is beneficial to construct the network using "cancer relevant edges" only and that for such networks the existence of background mutations is even helpful when combined with their diffusion based approach.

The application of their method returned several interesting results including stratification of head and neck cancer as well as pathways associated with poor survival in liver cancer and lung cancer.

Overall this is an interesting paper, however there are two issues that the authors should address before the paper could be published:

First, there is a significant disconnect between the simulated scenario and reality. The second criticism is minor and asks for softening the wording from claiming causality to reporting association (or strengthening the arguments to prove causality - I would be fine with either solution.)

Issues related to simulations:

There is a significant disconnect between simulated and real scenarios. Based on the authors' argument, frequently mutated pathways, FMPs, are subgraphs of the graph defined by cancer "relevant" interactions. At the same time "relevant" interactions are defined as protein-protein interactions supported by mutual exclusivity. However, in practice, ME graphs are sparse, star-like, with a highly mutated gene at the center of these stars. So there is a disconnect between this observation and the assumption about the properties of simulated FMPs.

In addition, there is a disconnect between the way the author models cancer non-relevant edges and their motivation. The authors started by arguing that cancer non-relevant edges make up much of the currently available PPI networks which, in turn, was the reason for replacing these networks with a subset of edges. Yet the authors model cancer non-relevant edges as random. This is in contradiction to the fact that even after removing ME supported edges, remaining PPI interactions are unlikely to be random. Instead, they most likely still maintain functionally related modules and adding such edges could be far less destructive than adding random edges. So this argument does not support the claim that it is better to use ESP than PPI.

Finally, the statement (first paragraph of the Results section) needs clarification: "In mutating FMPs mutations are assigned to a fixed number of pathway genes"; how exactly? It is not clear if and how mutual exclusivity is included in the modelling?

Issue related to association versus causation.

The authors show examples that mutations in certain subnetworks can be used to distinguish aggressive subtypes. However, rather than carefully stating that these mutations are "associated" with bad prognosis, they make a much stronger assessment that mutations in these pathways actually "drive" these subtypes. I looked carefully at the TP53-AXIN-ARHGEF17 pathway and, as it stands, the authors do not have sufficient support for claiming causality. Specifically, I would like to point out that ARHGEF17 is a known cancer gene - Tumor Endothelial Marker 4 (TEM4). TEMs are

antigens enriched in tumor versus non malignant endothelia. TEM4 expression is associated with bad prognosis. In addition this a rather long protein. This opens an alternative explanation for the association of TEM4 mutations with bad prognosis. Specifically, if a high expression of a gene is associated with prognosis this might lead to transcription induced mutations which, given the length of TEM4 gene, might be frequent enough to cause a correlation of this specific prognosis with mutations in TEM4. While it is possible to perform further analyses to support or refute such alternative explanation (e.g. showing that when controlling for expression differences mutations still provide prognostic information) my point is that it is possible for mutations in specific genes to be markers of cancer related events, including survival, without driving them and thus the authors should not rush into the "driver" conclusion.

Reviewer #3 (Remarks to the Author):

The authors have developed a novel way of subtyping cancer tumor types by utilizing a heat diffusion (network propagation) method through a carefully subsetted protein-protein interaction network which they call evolutionary selected pathway (ESP) map. The subsetting has been done by selecting biophysically relevant interactions from what authors call irrelevant interactions, as well as identifying pairs of genes that are mutually exclusively mutated across tumors in at least one tissue.

The authors show that a smaller and more biologically relevant protein-protein interaction network, the ESP map, can stratify cancer patients into subtypes and help in understanding heterogeneity of cancers in general. Presented results prove the need for high quality tumor-specific and cell- or tissue-specific data.

The idea of ranking the quality of PPI edges is not by itself novel, but the biophysical relevant interactions and mutually exclusively mutated genes across tumors is a novel and clever way to subset PPI networks and likely will be of high interest to other scientists in the field.

The claims are backed up by two study cases where authors were able to connect poor survival with drug resistance.

The authors thoroughly compared their approach to existing tools and databases, and they have made their code available at github.

The manuscript is well written and requires minor corrections prior publishing, particularly the following specific issues should be addressed:

Paragraph 3 (when the reference network ...): Please define what "clean" network means.

A TP53-AXIN-ARHGEF17 pathway driving aggressive liver cancer: Please state clearly which findings are novel and which reproduce earlier studies.

In online methods:

Explanation of equation 3 lists U two times, and does not explain transposed V. Possibly one of the U should be V or V^T instead?

Please clarify whether evolutionarily selected pathways are the collection of mutually exclusive genes, or if the mutually exclusive genes are a subset of the ESP.

Evolutionarily selected pathway construction: Not clear what values were used to calculate P values.

Subtypes driven by mutations of FMPs: Why 25 genes per pathway?

We thank all the reviewers for the very helpful comments. Here we summarize our modifications and responses to the reviewers' major concerns. We do not mention minor requested corrections such as typos. We highlighted all the major modifications in the paper.

Reviewer #1 (Remarks to the Author):

Question 1: Several aspect of this manuscript made it frustrating to review. The supposed rationale of this approach was that current approaches use protein-protein interactions that were uncovered in unrelated tissues or try to merge mutations data across cancer types. But then they go and use a protein-protein interaction dataset that is also not from a relevant cell type (they use the same protein-protein data for all cancer types). So, these protein-protein interactions are useful?

Answer: To our best knowledge, currently there are no such large-scale protein interaction data available that are tissue- or cancer-specific. That is exactly the challenge of our study -- to determine whether it is nonetheless possible to use existing protein interaction datasets by filtering them with outside information. In our study, we introduce one way to determine whether it is nonetheless possible to use existing protein interaction datasets but by filtering them with extra information. Similar strategies has been carried out successfully before, in other contexts. Previous examples in which various tissue-specific information, such as gene functional annotation and mRNA expression, has been applied to filter protein interaction networks are as follows: 1) Magger et al., PLoS Comput. Biol., 8, e1002690. 2012, 2) Guan et al., PLoS Comput. Biol., 8, e1002694, 2012. 3) Zitnik et al., Bioinformatics, 33, i190–i198., 2017, 4) Greene et al., Nat. Genet., 47, 569–576., 2015. These works are based on the assumption that a pair of proteins documented in the public protein interaction databases interact with each other if two proteins are both expressed or perform similar functions in a particular cell type. Based on their successes, we adopted a similar rationale to filter protein networks in cancer. Instead of using expression or gene functional information, we selected protein interactions based on the mutual exclusivity patterns of their encoding genes. Rather than simply using all interactions detected in various tissues and conditions, the rationale of our work was to select interactions that more possible to play a critical role in evolution and selection of mutations in the cancer genome. We added the corresponding text to address this issue in the last paragraph of introduction, as follows:

Ideally, such pathway analyses should rely on the specific molecular interactions that drive cancer in relevant tissue types, as opposed to interactions important for other cellular states, diseases and/or tissues. However, most types of experimental data used to inform molecular interaction networks, including protein-protein interactions and genetic interactions, cannot yet be readily generated at the scale necessary to cover many specific tumor samples or tissues.

Therefore, in nearly all cancer pathway analyses, molecular interaction information is drawn heavily from network meta-resources^{8-10,45}. These meta-resources are large, cataloging in the range of 10^3 - 10^7 interactions, as well as non-discriminatory, representing many diverse experiments in different human cell lines, primary tissues, or *ex-vivo* contexts such as yeast two-hybrid⁴⁶, with each source influenced by different rates of false-positive and false-negative errors.

While these meta-resources have been extremely useful, the high diversity of their contents motivates at least two major directions for further bioinformatics research. First, the effects of large numbers of non-specific interactions are not yet well understood. Is their inclusion in cancer pathway analyses helpful, neutral, or harmful? Second, it is not yet clear how to formulate molecular interaction networks that are both cancer-relevant and tissue-type specific. While various computational methods have been proposed to address tissue specificity, for instance by selecting interactions with tissue-specific gene expression patterns or functional annotations^{21,47-49}, similar strategies have not been devised for nominating interactions specific or relevant to cancer.

Question 2: Figure 2 was confusing, why are some interactions boxed with the cancer name. Presumably many of those interactions were found in more than one cancer type. Even for the colored lines, if these indicate a cancer in which mutually exclusivity was observed shouldn't some interactions have two or more colored lines?

Answer: We used boxes with cancer type labels to include the genes and interactions that are mutually exclusive in one cancer type only. As the referee also noticed, many interactions are shared by more than one cancer type. This was shown in Figure 2 by using black colored edges, explained in the legend as follows:

Black edges indicate interactions for multiple cancer types (Supplementary Table 3). Boxes indicate interaction neighborhoods enriched in specific cancer types.

We now include a separate **Supplementary Table 3** to report the cancer types affiliated with every edge.

Question 3: I am not sure the logic of looking at survival of ESP mut versus ESP nonmut patients. If mutation of the different genes in the ESP do essentially the same thing then it would have been more compelling to show that the gene expression states of tumors with each genes in the ESP look more like each than the ESP nonmut samples. Also, even if you did look at survival with the prediction that each gene in the ESP does that same thing, then the effect on survival should have also been the same and it wasn't.

Answer: Prompted by this reviewer suggestion, we investigated the relationship between ESP status (mutated or unmutated) and tumor expression profile (first principal component). Indeed, we found a significant number of ESPs in which mutation influences the expression state of the tumor. Moreover, these expression changes map at least partially to known cellular pathways associated with cancer hallmarks. We have added the following text and new results in our revision:

Main manuscript: We studied whether mutation of the identified ESPs leads to downstream effects on gene expression. Indeed, the alteration status (mutated or unmutated) of three ESPs, THCA_551, LUAD_531, and SARC_531, was associated with the first principal component of expression among tumor samples in thyroid carcinoma, lung, and sarcoma, respectively (**Supplementary Table 2**). More specifically, we also checked the influence of ESP alteration on the expression of known cellular pathways associated with cancer (**Online Methods**). In this case, we found a larger proportion of ESPs (82 out of 117) were associated with expression changes in at least one of these specific cancer pathways.

Online Methods: **Expression association.** We clustered tumors based on a selected ESP by using a random walk-based clustering procedure described in the previous section. We then evaluated whether the expression level of a certain set of genes was significantly different for ESP-mutated and ESP-non-mutated cohorts using a one tail ranksum test. The expression level of a set of genes was represented by the first principal component of the patient-by-gene expression matrix.

Question 4: While they uniquely uncover ESPs with their approach are these genes really in the same pathways and relevant to cancer? What data (other than survival which can be driven by many things) do they provide that they have found any "real" interactions?

Answer: We feel that our response to Question 3 above, establishing a link between genetic alterations in ESPs and changes in gene expression, also provide strong evidence to address this Question 4. Furthermore, the manuscript also includes results that link ESP mutation to drug responses.

Question 5: Some references do not show they claim that they do, for example ref 4-7 references in introduction are not all about cancer data sets. In the discussion, they state "...the biophysical interaction between STK11 and STK11IP had not been directly studies in cancer but

is known to play a role in PJS..." but the reference that is supposed to support this does not mention STK11IP. Clearly I am not going to check all of their references but they should learn that papers need to be references properly and you cannot try to support a claim with a reference that doesn't support it.

Answer: These references are indeed relevant, but we agree with the reviewer that the text could do a better job of explaining the rationale in some of these cases. We have now revisited all of the references in the paper and ensured that the rationale for each reference is clear from context. For example, as the review mentioned, for references mentioned above, we rephrased the text used in the paper from:

Such "pathway" analyses have been frequently applied to cancer datasets to identify disease mechanisms and biomarkers (ref 4-7)

To: **Such pathway analyses have been frequently applied to cancer datasets to aggregate gene-level signals to identify new pathway-level biomarkers (ref 4-7).**

Since the pathway we identified are novel, some interactions may not be well studied in previous cancer research papers but may be supported by research in other contexts. For example, regarding the interaction between STK11 and STK11IP mentioned by the reviewer, we stand by our earlier statement in the manuscript that these two proteins have indeed been found to interact in other diseases such as Peutz–Jeghers Syndrome. Ref 68 first states that these two proteins interact with other in this sentence: "*LKB1 has been suggested to mediate its cellular functions through interactions with a number of proteins including LIP1, BRG1, MO25a and STRADa*". We note that LKB1 is a synonym of STK11 and LIP1 is a synonym of STK11P. Here we simply demonstrated that there exist supporting evidence for our pathway analysis which may potentially be useful for readers to follow up.

Question 6: Many terms that they used were at least somewhat wrong which made the manuscript frustrating to read, For ex "18 tissue cohorts", "subtypes of a tissue" "within one or more tumor tissues". These are all from one page.

Answer: We are not entirely sure we understand the referee's point, but we have rewritten all of the mentioned phrases.

Question 7: Some figures are also labelled in a way that makes them more confusing than they need to be. Example, Figure 6d why does that say ESP-LIHC on the top when it is ESP as well as each gene separately?

Answer: We have removed "ESP" from the title of each panel in **Figure 6**.

Question 8: Figure 6e for the ESP mutated, what does that look like for each single genes (for example p53 mutant tumor would be expected to be resistant). For 6D+I they make the point

that the ESP is better than each single gene but then for 6E+J then don't show the data for each individual gene.

Answer: We have added the results showing that individual genes cannot achieve better drug responses than ESPs. These new results are in **Supplementary Figure 14** as follows:

Supplementary Figure 14. ESP stratification of drug resistance phenotype in liver (LIHC) and lung (LUAD) tumors. a, Drug resistance (log IC50) on cell-line data using the aggregate of genes in the liver ESP versus each of its genes individually. **b,** Drug resistance (log IC50) on cell-line data using the aggregate of genes in the lung ESP versus each of its genes individually.

Question 9: For the individual genes shown in 6D and 6I isn't the p-value affected the number of samples? What is the HR for each comparison.

Answer: We agree with the reviewer. The *P*-value is indeed affected by the number of samples. We have now additionally provided the Hazard Ratio for each comparison as suggested by the reviewer. We find that, generally, the HR of ESP biomarkers is not more pronounced than the HR of individual gene biomarkers. Rather, the benefit of the ESP is to diagnose a greater number of patients than would be indicated by any of its member genes individually. The p-value does capture this benefit, thus we have elected to show both the p-value and the HR in the revised figure and add clarifying text to the revised manuscript.

Reviewer #2 (Remarks to the Author):

Question 1: There is a significant disconnect between simulated and real scenarios. Based on the authors argument, frequently mutated pathways, FMPs, are subgraphs of the graph defined by cancer “relevant” interactions. At the same time “relevant” interactions are defined as protein-protein interactions supported by mutual exclusivity. However, in practice, ME graphs are sparse, star-like, with a highly mutated genes at the centers of these stars. So there is a disconnect between this observation and the assumption about the properties of simulated FMPs. In addition, there is a disconnect between the way the author model cancer non-relevant edges and their motivation. The authors started by arguing that cancer non-relevant edges make much of the currently available PPI networks which, in turn was the reason of replacing these networks with a subset of edges. Yet the authors model cancer non-relevant edges as random. This is in contradiction to the fact even after removing ME supported edges, remaining PPI interactions are unlikely to be random. Instead they most likely still maintain functionally related modules and adding such edges could be far less destructive than adding random edges. So this argument does not support the claim that it is better to use ESP than PPI.

Answer: We have now conducted further simulation experiments along the lines suggested by the reviewer. In the new simulations, FMPs are sparse and star-like and the node degree distribution follows a power law to mimic real PPI networks. Notably, we observe similar trends to those in the original simulations of the paper. We now include the new figure and corresponding text as follows:

Texts added to the main text:

Random networks were simulated using an Erdos-Renyi model. We found empirically that the method for generating random interactions, preferential attachment or Erdos-Renyi, did not have a large effect on the analysis (Supplementary Figs. 2-3).

Texts added to the **Online Methods**:

We consider two types of network structure regarding the different structures of FMP and random interactions. In our first setting, we set FMP as densely connected subnetworks with various edge densities (from 0% to 100%) and uniformly sample m (from 0% to 2.5%) random edges using an Erdos-Renyi model. In the second setting, FMP is a star graph and random edges are samples using preferential attachment model.

Figures added to the supplementary:

Supplementary Figure 2. Exploring cancer pathway analysis through simulation. The simulated network contains star-like FMPs and random interactions generated by the preferential attachment model (see **Online Methods**). **a**, Tumor stratification performance with increasing frequency of background mutations, when no random interactions are presented. Performance is measured by calculating Adjusted Random Index (ARI) between the true subtypes and the tumor clusters derived by network stratification. **b**, Tumor stratification performance with increasing random interaction density.

Supplementary Figure 3. Accuracy landscape for different simulations. The simulated network contains star-like FMPs and random interactions generated by the preferential attachment model (see **Online Methods**). **a**, ARI landscape across varying background mutation frequency and density of random interactions when subtypes are driven by mutations of an FMP. **b**, ARI landscape across varying background mutation frequency and density of random interactions when subtypes are driven by an FMG. **c**, ARI landscape across varying background mutation frequency and density of random gene interactions when subtypes are driven by mutations of an FMP without any FMG present.

Question 2: Finally, the statement (first paragraph of the Results section) needs clarification: “In mutating FMPs, mutations are assigned to a fixed number of pathway genes”; how exactly? It is not clear if and how mutual exclusivity is included in the modelling?

Answer: In the **Online Methods**, we now describe these details as: “Each subtype was assigned with 25 genes in the pathway. The density of interactions within each pathway was set to 0.2 by default. For each tumor, there were two random genes mutated within each pathway.” We have now added a sentence at the end of second paragraph of the Results section to direct the readers to Online Methods as follows:

For further details of the simulations, see **Online Methods**.

Question 3: The authors show examples that mutations in certain subnetworks can be used to distinguish aggressive subtypes. However rather than carefully stating that these mutations are “associated” with bad prognosis they make a much stronger assessment that mutations in these pathways actually “drive” these subtypes. I looked carefully at the TP53-AXIN-ARHGEF17 pathway and, as it stand, the authors do not have sufficient support for claiming causality. Specifically I would like to point out that ARHGEF17 is a known cancer gene - Tumor Endothelial Marker 4 (TEM4). TEMs are antigens enriched in tumor versus non malignant endothelia. TEM4 expression is associated with bad prognosis. In addition this a rather long protein. This opens an alternative explanation for the association of TEM4 mutations with bad prognosis. Specifically, if a high expression of a gene is associated with prognosis this might lead to transcription induced mutations which, given the length of TEM4 gene, might be frequent enough to cause a correlation of this specific prognosis with mutations in TEM4. While it is possible to perform further analyses to support or refute such alternative explanation (e.g. showing that when controlling for expression differences mutations still provide prognostic information) my point is that it is possible for mutations in specific genes to be markers of cancer related events, including survival, without driving them and thus the authors should not rush into the “driver” conclusion.

Answer: We agree with the reviewer’s suggestion that it is too early to reach the conclusion that our pathways drive the cancer subtype. Therefore, we have softened the title of the pathway section to “A TP53-AXIN-ARHGEF17 pathway **associated** with aggressive liver cancer”.

Reviewer #3 (Remarks to the Author):

Question 1: Paragraph 3 (when the reference network ...): Please define what "clean" network means.

Answer: A “clean” network means there are no cancer irrelevant (random) protein interactions in the network. We modified this sentence to make it more clear in the paper, as follows:

When the reference network contained few random interactions (<1%), we found that FMP-driven subtypes were recovered with very high accuracy, even in the presence of background mutations.

Question 2: A TP53-AXIN-ARHGEF17 pathway driving aggressive liver cancer: Please state clearly which findings are novel and which reproduce earlier studies.

Answer: As suggested, we have revised the text for this section to clarify, as follows:

We identified two interesting properties that had not been reported by previous studies relating to genes in this ESP. First, mutating this ESP indicated low survival, which could be further validated in a second large cohort of liver tumors from ICGC, supporting the value of this ESP as a clinical biomarker (Hazard ratio 2.7, Fig. 6c). Notably, none of the three genes involved in this ESP (TP53, AXIN1, and ARHGEF17) were strongly prognostic when mutations to each gene were considered individually (Fig. 6d). Second, given that the AXIN1-TP53 interaction modulates DNA damage response, we also tested associations between this ESP and DNA damaging chemotherapeutics widely used to treat liver cancer (Online Methods). This investigation was carried out in a panel of 19 liver cancer cell lines characterized in the Genomics of Drug Sensitivity in Cancer (GDSC) dataset, of which 15 had mutations placing them in the ESP subtype. This analysis showed that ESP subtype mutations are indeed associated with strong resistance to mitomycin C (Fig. 6e), consistent with the observation of poor survival in both TCGA and ICGC (Figs. 6b,c).

Question 3: Explanation of equation 3 lists U two times, and does not explain transposed V. Possibly one of the U should be V or V^T instead?

Answer: We thank the reviewer for recognizing this typographical error. The second U should be V. We have corrected it in the new version.

Question 4: Please clarify whether evolutionarily selected pathways are the collection of mutually exclusive genes, or if the mutually exclusive genes are a subset of the ESP.

Answer: An ESP is defined as a collection of gene-gene interactions, in which each interaction is supported by a biophysical interaction between the gene products (i.e. a protein-protein interaction) and mutual exclusivity of somatic mutations to the two genes. We have added the corresponding text at the end of the Introduction, as follows:

We found that by integration of these interactions with protein-protein binding data, we could successfully detect biologically and clinically informative cancer subtypes. We therefore define these subnetworks of mutually exclusive and physical interactions as Evolutionarily Selected Pathways (ESP).

Question 5: Evolutionarily selected pathway construction: Not clear what values were used to calculate P values.

Answer: The mutually exclusive P value for two genes is calculated using their mutation profile. We now make that statement explicit in the paper:

For each cancer type, we selected the top K ($=100$) most significant mutually exclusive mutated gene pairs ranked by P -value. In particular, for any of two genes, we calculated the number of patients for whom 1) both genes are mutated; 2) the first gene is mutated; 3) the second gene is mutated; 4) neither of the two genes is mutated. We then calculated the P -value using a one tailed Fisher's exact test with these four numbers.

Question 6: Subtypes driven by mutations of FMPs: Why 25 genes per pathway?

Answer: There are three factors determining the contribution of an FMP to the patient clustering: (1) number of genes within each pathway; (2) number of mutations within each pathway; (3) density of gene interactions within each pathway. In the simulation, in order to evaluate the impacts of FMGs and FMPs to the patient clustering, it is not necessary to examine all combinations of these three factors. For example, increasing the number of genes or the ratio of gene interactions will both bias the clustering results towards FMPs. Therefore, for clarity, we simply fixed factors (1) (number of genes = 25) and (2) and enumerated factor (3) as shown in **Supplementary. Figure 1**.

REVIEWERS' COMMENTS:

Reviewer #1 (Remarks to the Author):

The authors have addressed most of my comments in a reasonable manner.

Reviewer #2 (Remarks to the Author):

The authors have addressed my concerns.

We thank all the reviewers for the very helpful comments.

REVIEWERS' COMMENTS:

Reviewer #1 (Remarks to the Author):

The authors have addressed most of my comments in a reasonable manner.

Reviewer #2 (Remarks to the Author):

The authors have addressed my concerns.